# Epitaxial Growth of Sc_0.09_Al_0.91_N and Sc_0.18_Al_0.82_N Thin Films on Sapphire Substrates by Magnetron Sputtering for Surface Acoustic Waves Applications

**DOI:** 10.3390/s20164630

**Published:** 2020-08-17

**Authors:** Florian Bartoli, Jérémy Streque, Jaafar Ghanbaja, Philippe Pigeat, Pascal Boulet, Sami Hage-Ali, Natalya Naumenko, A. Redjaïmia, Thierry Aubert, Omar Elmazria

**Affiliations:** 1Institut Jean Lamour, UMR 7198 Université de Lorraine–CNRS, 54000 Nancy, France; florian.bartoli@univ-lorraine.fr (F.B.); jeremy.streque@gmail.com (J.S.); jaafar.ghanbaja@univ-lorraine.fr (J.G.); philippe.pigeat@univ-lorraine.fr (P.P.); p.boulet@univ-lorraine.fr (P.B.); sami.hage-ali@univ-lorraine.fr (S.H.-A.); abdelkrim.redjaimia@univ-lorraine.fr (A.R.);; 2National University of Science and Technology “MISIS”, 119049 Moscow, Russia; nnaumenko@ieee.org; 3Laboratoire Matériaux Optiques, Photonique et Systèmes (LMOPS), Université de Lorraine–CentraleSupélec, 57070 Metz, France

**Keywords:** ScAlN thin films, sputtering, hetero-epitaxial films, SAW sensors

## Abstract

Scandium aluminum nitride (Sc_x_Al_1−x_N) films are currently intensively studied for surface acoustic waves (SAW) filters and sensors applications, because of the excellent trade-off they present between high SAW velocity, large piezoelectric properties and wide bandgap for the intermediate compositions with an Sc content between 10 and 20%. In this paper, the growth of Sc_0.09_Al_0.91_N and Sc_0.18_Al_0.82_N films on sapphire substrates by sputtering method is investigated. The plasma parameters were optimized, according to the film composition, in order to obtain highly-oriented films. X-ray diffraction rocking-curve measurements show a full width at half maximum below 1.5°. Moreover, high-resolution transmission electron microscopy investigations reveal the epitaxial nature of the growth. Electrical characterizations of the Sc_0.09_Al_0.91_N/sapphire-based SAW devices show three identified modes. Numerical investigations demonstrate that the intermediate compositions between 10 and 20% of scandium allow for the achievement of SAW devices with an electromechanical coupling coefficient up to 2%, provided the film is combined with electrodes constituted by a metal with a high density.

## 1. Introduction

In 2009, Akiyama et al. experimentally demonstrated that scandium aluminum nitride (Sc_x_Al_1−x_N) thin films show a large enhancement of the d_33_ piezoelectric constant as the scandium content increases up to 43%: the d_33_ coefficient of Sc_0.43_Al_0.57_N films is more than four times larger than that of aluminum nitride (AlN) [1,2]. It has been shown that this phenomenon is related to the decrease of the C_33_ elastic constant [3]. This result opened exciting new perspectives related to the enhancement of the electromechanical coupling coefficient k^2^ of high-frequency acoustic waves devices based on AlN material. This phenomenon has been experimentally proved in the case of surface acoustic waves (SAW) devices [4,5], bulk acoustic waves (BAW) devices [6,7] or Lamb waves resonators (LWR) [8]. For instance k^2^ values close to 5% have been reached for SAW devices [5], and 12.8% for BAW devices [7], while a factor of merit of 18 has been achieved for LWR [8], making Sc_x_Al_1−x_N thin films serious candidates for 5G filters applications [6].

This improvement in k^2^ could also be of great interest for another important field of application of SAW devices, namely high-temperature sensing. As they are passive devices, SAW sensors offer attractive prospects for remote monitoring and control of moving parts, especially at high temperature, i.e., above 200 °C and potentially up to 800 °C and more [9]. This technology requires the use of a piezoelectric material able to withstand these very harsh conditions. In this context, langasite (La_3_Ga_5_SiO_14_; LGS) crystals have been demonstrated to be very serious candidates. Indeed, langasite crystals are exceptionally stable at high temperatures, showing no phase transition up to their melting temperature at 1470 °C [10]. This outstanding property allowed the achievement of wireless SAW resonators able to be operated at 330 MHz up to 750 °C for a few hours [11]. Higher levels of performances are limited by several factors. First, the electrical resistivity of langasite crystals drops dramatically with the temperature, to reach only 10^6^ Ω·cm at 600 °C, which induces leakage currents between the electrode fingers and thus reduces the performances of langasite-based SAW sensors at higher temperatures [12]. Furthermore, acoustic propagation losses in langasite strongly increase with frequency at high temperature, which prevents its use in the gigahertz range. Finally, its moderate k^2^, less than 0.5%, is not compatible with the reflective delay lines configuration, which is particularly interesting, as it enables the sensor identification via the time domain response of the sensor, which is like a binary code resolved in time [13]. In the last decade, the AlN/sapphire bilayer structure has been identified as a potential candidate to overcome some of the langasite limitations. Indeed, AlN also shows a remarkable high-temperature stability, at least up to 1000 °C, like many III-N materials [14]. Moreover, because of its wide bandgap of 6.2 eV, AlN has outstanding dielectric properties, superior to those of langasite by several orders of magnitudes at any temperature [15]. In addition, SAW velocities in AlN are close to 5500 m/s, making this material suitable for applications in the most promising industrial, scientific, and medical (ISM) 2.45 GHz band [16]. In this context, sapphire substrates are highly compatible with AlN films as they show concurrently high temperature stability, high dielectric properties, comparable SAW velocities, and they enable the growth of highly textured AlN films [17]. Finally, the AlN/sapphire structure exhibits relatively low propagation losses at high frequencies [18]. One major result concerning the use of the AlN/sapphire bilayer structure for high-temperature SAW applications consisted of the measurement of a SAW signal on a wired delay line for 40 h at 1050 °C [19]. However, despite this promising result, AlN/sapphire is also seriously limited by a moderate k^2^ value around 0.3% [20]. This drawback could be overcome by the replacement of AlN by Sc_x_Al_1−x_N films, which provide a unique tradeoff between strong electromechanical properties (k^2^ ≈ 1%) and high-temperature compatibility [1]. Indeed, the piezoelectric properties of Sc_x_Al_1−x_N films have been proved to be stable up to 1000 °C independently from the Sc rate [21]. Furthermore, it has been experimentally shown that the electromechanical coupling factor of BAW resonators based on Sc_x_Al_1−x_N films with a Sc atomic content from 7 to 10%, increases by 20% after a 15-min annealing period at 600 °C under vacuum [22]. Moreover, the elastic constants, and thus the SAW velocity of Sc_x_Al_1−x_N films decrease with the Sc quantity, reaching 4000 m/s for a Sc content of 40% [23]. This value is still compatible with the 2.45 GHz ISM band provided the films are grown on a high velocity substrate such as sapphire. Additionally, this combination of a slow film over a fast substrate might give birth to high-velocity higher-order modes. The main point of attention regarding high-temperature SAW applications of Sc_x_Al_1−x_N films is related to the decrease of the bandgap with the scandium content, reaching less than 3 eV for a Sc content of 40% [24]. The use of intermediate compositions between 10 and 20% of scandium could be in this regard particularly interesting. Table 1 summarizes the properties of LGS, AlN/sapphire and Sc_0.09_Al_0.91_N/sapphire materials.

In order to develop in the future a passive high-temperature SAW sensor based on the Sc_x_Al_1−x_N/sapphire structure, it is necessary to minimize all sources of losses. In particular, the crystalline quality of the Sc_x_Al_1−x_N film must be optimized to enhance its piezoelectric properties. This work has been started by many research groups, studying the influence of sputtering process parameters on the crystalline quality of the films. Parameters such as substrate temperature [25], gas pressure, discharge power [26] and gas ratio [27] have been investigated. The synthesis of Sc_x_Al_1−x_N films by other techniques, such as molecular beam epitaxy (MBE) [28] and metal organic chemical vapor deposition (MOCVD) [29], has also been studied.

In this paper, the growth of epitaxial Sc_x_Al_1−x_N thin films on sapphire substrates by reactive magnetron sputtering is described. Three Sc_x_Al_1−x_N compositions, respectively with x = 0, 0.09 and 0.18, are investigated. The strategy employed to find the optimized sputtering parameters for the Sc_x_Al_1−x_N films was to study firstly the composition with x = 0 (i.e., a pure AlN film), which is easier to grow. Once those parameters known for AlN, they were then slightly modified to find the new optimum sputtering parameters for the Sc_0.09_Al_0.91_N films. These new parameters were then adapted again for the growth of the Sc_0.18_Al_0.82_N films. Then SAW resonators were made on the prepared Sc_x_Al_1−x_N/sapphire samples and electrically characterized at room temperature. The behavior of the bilayer structure with high density electrodes, such as platinum electrodes, which show good chemical properties for high-temperature applications, has been numerically investigated.

## 2. Methods

500 nm-thick c-axis oriented AlN, Sc_0.09_Al_0.91_N, and Sc_0.18_Al_0.82_N thin films were deposited on (0006) sapphire substrates by cathodic reactive magnetron sputtering. The sputtering system is a non-commercial rf machine, which possesses one magnetron able to host 2 in. targets. A composite target made of 99.99% pure Sc and Al pieces was used. The dimensions and the number of pieces were adapted to obtain the targeted compositions (Figure 1). The films thicknesses were measured by in situ reflectometry, using a 400 nm wavelength laser.

Sapphire substrates as well as Al and Sc targets were purchased from Neyco Vacuum & Materials Company, Vanves, France. The substrate holder is not biased and can be heated by a tungsten heating resistance. The substrate temperature is controlled by a K-type thermocouple. The base pressure in the chamber is around 10^−6^ Pa. During the sputtering process, high-purity nitrogen (99.999%) gas is introduced. Several sputtering parameters, namely the substrate temperature and the plasma pressure, have been varied to obtain epitaxial Sc_x_Al_1−x_N thin films. These conditions are summarized in Table 2. Table 3 gives the investigated sputtering parameters for each of the three grown materials.

The microstructure was firstly determined by θ-2θ and rocking-curve X-ray diffraction (XRD) measurements, using a PANalytical X’Pert Pro MRD high-resolution diffractometer. The primary optic is a hybrid monochromator which allows a parallel monochromatic Cu Kα1 beam. As a secondary optic, there is a parallel plate collimator limiting the horizontal divergence to 0.27°. This diffractometer uses a PIXcel detector, used in punctual mode in our case.

Transmission electron microscopy (TEM) and scanning transmission electron microscopy (STEM) investigations were carried out using a JEM-ARM 200F Cold FEG TEM/STEM (JEOL Ltd., Tokyo, Japan), operating at 200 kV and equipped with a spherical aberration (Cs) probe and image correctors (point resolution 0.12 nm in TEM mode and 0.078 nm in STEM mode). Chemical compositions were determined using energy dispersive X-Ray spectroscopy (EDS). EDS spectra were recorded in STEM mode by means of a Centurio Jeol spectrometer (SDD). TEM lamellas were prepared by focused ion beam (FIB) method, using a FEI Helios Nanolab 600i (FEI, Hillsboro, Oregon, USA).

The selected area electron diffraction (SAED) patterns were indexed using the CrystalMaker-SingleCrysal^®^ software (CrystalMaker Software Ltd., Oxford, UK, www.crystalmaker.com).

The structure selected for the microfabrication of SAW resonators was a 2.76 µm-thick Sc_0.09_Al_0.91_N/sapphire bilayer structure. A 150 nm-thick aluminum films was sputtered onto the bilayer structure. One-port synchronous resonators were then patterned by conventional UV lithography, combined with wet etching of the aluminum film. SAW resonators with various wavelengths were achieved, including λ = 6.5 µm and λ = 13 µm, which are the typical values used on standard AlN/sapphire structure to address the 433 and 868 MHz ISM bands. These synchronous resonators have 100 finger pairs and 200 reflectors on each side, with a metallization ratio of 50%, and an aperture of 40·λ. SAW devices were characterized at room temperature, using a probe station Süss Microtech PM5 (Süss MicroTec Lithography GmbH, Sternenfels, Germany) and a VNA Agilent-N5230A network analyzer (Agilent Technologies, Santa Clara, CA, USA).

## 3. AlN and Sc_x_Al_1−x_N Growth and Characterization

The recorded XRD θ-2θ diagrams (Figure 2, Figure 3 and Figure 4) are consistent with a scandium aluminum nitride thin film deposited on a sapphire substrate (α-Al_2_O_3_). The crystal structures of α-Al_2_O_3_ and Sc_x_Al_1−x_N for Sc content below 43% [1], belong to $R\bar{3}c$ and $P6_{3}\mathrm{mc}$ space groups, respectively (Table 4). Although belonging to the trigonal system, α-Al_2_O_3_ is described using a hexagonal cell, just like Sc_x_Al_1−x_N [3]. It has to be noted that as Sc_x_Al_1−x_N is a relatively new material, crystallographic databases are not available for any composition. However, the lattice parameters for x = 0.09 or 0.18 are very close to AlN ones [1].

### 3.1. AlN as Reference for Sc_x_Al_1−x_N

The first study consisted in the deposition of AlN thin films on (0006) sapphire substrates. In that case, the pressure was kept constant at 5 mTorr, and only the substrate temperature was varied. Five different temperatures from the ambient to 560 °C were explored (Table 3). XRD θ-2θ measurements reveal the enhancement of the targeted (0002) orientation of AlN when the temperature rises from the ambient to 470 °C, while the parasitic peaks related to (1010) or (1120)-oriented grains slightly disappear. At 470 °C, only the (0002) and (0004) reflexes of AlN can be observed in the XRD spectrum (Figure 2). Above this temperature, the intensity of the (0002) peak decreases, while small peaks related to misoriented grains reappear. XRD rocking-curve measurements made on the (0002) peak exhibit a full width at half-maximum (FWHM) of 0.37° (Figure 2 inset), confirming the excellent out-of-plane texture of the films deposited at 470 °C. Therefore, the corresponding sputtering parameters set has been used as a starting point for the study of the Sc_x_Al_1−x_N films growth on sapphire substrate.

### 3.2. Growth of Sc_0.09_Al_0.91_N Thin Films

For this part of the study, the Sc content in the composite target was equal to 12.5% (Table 3). Nevertheless, EDS measurements showed that the actual Sc/Al ratio in the film is 8.8/91.2 (approximated to 9/91). This is related to the lower sputtering rate of scandium as compared to that of aluminum. When using the optimized sputtering parameters set deduced from the AlN deposition study (see hereinabove), the (0002) orientation of the Sc_0.09_Al_0.91_N films was kept as expected, but the texture of the film was slightly worse, as compared to that of the optimized AlN films. We attribute this phenomenon to the modification of the electrical conditions in the chamber, related to the deposition of a ScAlN layer on the inner wall of the sputtering machine (in place of an AlN layer) that induces an increase of the plasma tension (for a constant sputtering power of 200 W). Therefore, the energy of the sputtered species becomes higher. To compensate, the plasma pressure was elevated from 5 to 7 mTorr, in order to decrease the mean free path of these species (Table 3). We also assumed that the substitution of some Al atoms by heavier Sc atoms induces an increased need of energy for the atoms and molecules to move at the surface of the sapphire substrate. We decided to bring this energy by an increase of the temperature substrate. (Table 3).

Among the tested temperatures listed in Table 3, XRD θ-2θ measurements revealed that the optimum substrate temperature to manage a highly textured (0002) orientation for Sc_0.09_Al_0.91_N is 650 °C. The increase of the plasma pressure from 5 to 7 mTorr enables an additional improvement of the texture. However, the latter deteriorates if the temperature is increased over 650 °C, and if the plasma pressure is over 7 mTorr. Thus, the optimized films are obtained for a pressure of 7 mTorr and a substrate temperature of 650 °C. In that case, only the (0002) and (0004) reflexes of the Sc_0.09_Al_0.91_N films can be seen on the XRD θ-2θ diagram (Figure 3). Rocking-curve measurements achieved on the (0002) peak confirmed the high texture of the film, since the FWHM is equal to 0.78° (Figure 3 inset). This optimized sputtering parameters set was used as a starting point for the study of the growth of the next Sc_x_Al_1−x_N film composition.

### 3.3. Growth of Sc_0.18_Al_0.82_N Thin Films

For this part of the study, the Sc content in the composite target was doubled to 25% (Table 3). As expected, the actual Sc/Al ratio in the film, as revealed by EDXS measurements, was doubled as well, reaching 18/82. As in the case of the Sc_0.09_Al_0.91_N films deposition, a plasma tension increase was observed and been compensated by a pressure elevation, from 7 to 9 mTorr (Table 3). Moreover, it turned out that the texture of the film does not improve anymore if the temperature is increased over 650 °C. The films deposited at this last temperature and at a pressure of 9 mTorr are highly (0002)-oriented (Figure 4), as confirmed by the XRD rocking-curve measurements whose FWHM is equal to 1.2° (Figure 4 inset).

### 3.4. TEM Measurements of the Sc_0.18_Al_0.82_N Thin Film

Figure 5 shows the STEM-EDS mapping of the Sc_0.18_Al_0.82_N film for nitrogen (green), aluminum (red) and scandium (yellow). The K_α_ spectral line has been used for all characterized chemical elements. As it can be seen, at the nanoscale level, it is not possible to observe any chemical inhomogeneity in the Sc_0.18_Al_0.82_N thin film.

The TEM bright-field images show that the Sc_0.18_Al_0.82_N thin film exhibits a columnar microstructure (Figure 6a). The column width of the Sc_0.18_Al_0.82_N film is 175 nm on average. The SAED analysis recorded from the area around the interface between the Sc_0.18_Al_0.82_N thin film, and the sapphire substrate shows only distinct spots (Figure 6b). This result is consistent with the highly textured behavior observed by XRD analysis. An important surface roughness between 10 and 30 nm is observed. This level of roughness being hardly compatible with the conventional photolithography process, further experiments on the sputtering process are ongoing to achieve Sc_0.18_Al_0.82_N films with a smoother surface. The TEM bright-field image shows a film thickness around 500 nm. The composite SAED pattern has been recorded along the ${\left[1\bar{2}\bar{1}0\right]}_{Sc_{0.18}Al_{0.82}N}∥{\left[2\bar{1}\bar{1}0\right]}_{Al_{2}O_{3}}$ zone axis (Figure 6b), from the area on either side of the interface between the Sc_0.18_Al_0.82_N thin film and the substrate (Figure 6a). The corresponding simulated diffraction pattern (Figure 6c) reveals the orientation relationship developed between the deposited thin film and the substrate, as follows:$${\left(0006\right)}_{Al_{2}O_{3}}\;//\;{\left(0002\right)}_{Sc_{0.18}Al_{0.82}N}$$
$${\left(0\bar{3}30\right)}_{Al_{2}O_{3}}\;//\;{\left(\bar{1}\bar{1}20\right)}_{Sc_{0.18}Al_{0.82}N}$$
$${\left[2\bar{1}\bar{1}0\right]}_{Al_{2}O_{3}}\;//\;{\left[\bar{1}2\bar{1}0\right]}_{Sc_{0.18}Al_{0.82}N}$$

The TEM image and the corresponding SAED patterns point out that the growth direction for the Sc_0.18_Al_0.82_N is parallel to ${\left[0001\right]}_{Al_{2}O_{3}}\;//\;{\left[0001\right]}_{Sc_{0.18}Al_{0.82}N}$ while the interface is parallel to ${\left(0006\right)}_{Al_{2}O_{3}}\;//\;{\left(0002\right)}_{Sc_{0.18}Al_{0.82}N}$ planes.

High resolution TEM (HRTEM) micrographs recorded from the Sc_0.18_Al_0.82_N/sapphire samples show that the interface between the film and the substrate is well defined, being only 1 to 2 nm-thick, corresponding to three or four atomic layers (Figure 7a). The fast Fourier transform (FFT) pattern obtained from the Sc_0.18_Al_0.82_N film (Figure 7b) shows only discrete spots, which confirms the highly textured behavior of the film as well.

The FFT recorded from the interface between the film and the substrate (Figure 7c) shows a composite reciprocal lattice from the two phases. The indexing of spots observed on the FFT for both Sc_0.18_Al_0.82_N and sapphire (Figure 7b,d) are in agreement with those obtained from the SAED pattern (Figure 6b).

## 4. SAW Characterization Results and Numerical Simulation

The achieved devices were electrically characterized at room temperature. Three peaks corresponding to three different SAW modes are visible on the obtained spectra, located at 720 MHz, 930 MHz, and 1530 MHz respectively (Figure 8). A previous work, including a simulation study, revealed that the three observed modes respectively correspond to a Rayleigh SAW, a shear horizontal (SH) SAW, and a high-velocity longitudinal leaky SAW. The latter can show low attenuation coefficients if a careful choice of the film thickness, as well as the electrode nature and thickness, is made [30]. The experimental velocities of these three modes are of 4680 ms^−1^, 6045 ms^−1^ and 9945 ms^−1^ respectively. These results show that the Sc_x_Al_1−x_N/sapphire structure with a Sc rate lower than 20% has a strong potential for SAW applications, if k^2^ values significantly higher than that of the AlN/sapphire structure can be achieved.

The electromechanical coupling k^2^ of SAW modes propagating in Sc_x_Al_1−x_N/sapphire structure generally increases with the Sc_x_Al_1−x_N thickness, due to the better confinement of displacements in the piezoelectric film. In a structure with a fixed Sc_x_Al_1−x_N thickness, higher k^2^ can be achieved if the Sc content grows [30]. The coupling of the Rayleigh SAW also depends on the wave structure: with increasing the ratio u_3_/u_1_, where u_1_ and u_3_ are the tangential and vertical motions respectively, accompanying the wave propagation in the film, k^2^ also grows because the electrostatic potential is coupled only with u_3_. The transformation of the wave structure required to achieve a higher k^2^ can be obtained, for example, by the variation of the electrode thickness. Figure 9 shows the dependence of the Rayleigh SAW velocity and k^2^ on the electrode thickness in the Sc_0.18_Al_0.82_N/sapphire structure with Al, Cu or Pt grating. The thickness of the Sc_0.18_Al_0.82_N film is 0.425λ. The effective SAW velocity was estimated at the resonant frequency of the SAW resonators arranged on top of the bilayer structure and extracted from numerical admittance functions. These functions were computed using the numerical technique SDA-FEM-SDA [31,32], combining the finite element modeling (FEM) analysis of electrodes with the spectral-domain analysis (SDA) of a layered substrate. The electromechanical coupling k^2^ was estimated from the extracted difference between resonant and anti-resonant frequencies. The same numerical technique was previously used to explain the nature of the acoustic modes observed experimentally and predict the existence of low-attenuated high-velocity longitudinal leaky SAWs in Sc_x_Al_1−x_N/sapphire with optimized film and electrode thicknesses [30].

With increasing mass load of the grating caused by thicker electrodes, or electrode metals with a higher density, the SAW velocity decreases while the wave structure transforms, the ratio u_3_/u_1_ grows and k^2^ also grows. For instance, k^2^ = 0.4% can be achieved with Al electrodes having a thickness of 0.1λ. The same value of k^2^ can be achieved with thinner (0.032λ) Cu electrodes or even much thinner Pt electrodes (0.015λ). For comparison, Rayleigh SAW characteristics were also estimated for the Sc_0.4_Al_0.6_N/sapphire structure with Al grating and it revealed that higher electromechanical coupling values can be achieved in a structure with a Sc rate of only 18% and Pt electrodes. The Pt/Sc_0.18_Al_0.82_N/sapphire SAW structure enables the generation and propagation of a Rayleigh mode with k^2^ values higher than 1% when the electrodes thickness is over 0.037λ (which corresponds to 240 nm for a wavelength of 6.5 µm), and up to 2% when the electrode thickness reaches 0.1λ.

## 5. Conclusions

In this paper, the growth of epitaxial AlN, Sc_0.09_Al_0.91_N and Sc_0.18_Al_0.82_N thin films on sapphire substrates has been investigated. It has been established, through a TEM study, that the growth direction for the Sc_0.18_Al_0.82_N is parallel to the ${\left[0001\right]}_{Al_{2}O_{3}}\;//\;{\left[0001\right]}_{Sc_{0.18}Al_{0.82}N}$ direction, while the film/substrate interface is parallel to the ${\left(0006\right)}_{Al_{2}O_{3}}\;//\;{\left(0002\right)}_{Sc_{0.18}Al_{0.82}N}$ planes. It was shown that the addition of scandium in the film leads to an increase of the needed energy to optimize the atoms mobility at the surface of the substrate. This extra energy was brought by raising the substrate temperature. It was also observed that the deposition of different Sc_x_Al_1−x_N compositions on the inner walls of the sputtering machine increases the plasma tension. This phenomenon has been successfully compensated for by increasing the plasma pressure. With these adjustments, the XRD (0002) reflections show FWHM rocking-curves values of 0.37°, 0.78° and 1.2° for AlN, Sc_0.09_Al_0.91_N and Sc_0.18_Al_0.82_N thin films, respectively. EDS characterizations highlights the excellent chemical homogeneity of the films, which validates the use of a composite target for the deposition. SAW devices were made on Sc_0.09_Al_0.91_N/sapphire bilayer structures and successfully characterized. Three SAW modes were observed: a Rayleigh mode, a shear horizontal mode, and a longitudinal leaky mode. These SAW measurements were further studied in our previous work [30]. Though electromechanical coupling of SAW modes in the Sc_x_Al_1−x_N/sapphire bilayer structure generally grows when the Sc content increases up to 40%, the numerical investigation of the coupling coefficient as a function of the electrode metal and thickness, as well as the Sc content, revealed that a trade-off between the degradation of the dielectric properties of ScAlN and the improvement of its piezoelectric properties with Sc content can be achieved, if the intermediate composition between 10 and 20% of scandium is combined with electrode metal with a high density, like Pt for instance.

## Figures and Tables

**Figure 1 sensors-20-04630-f001:**
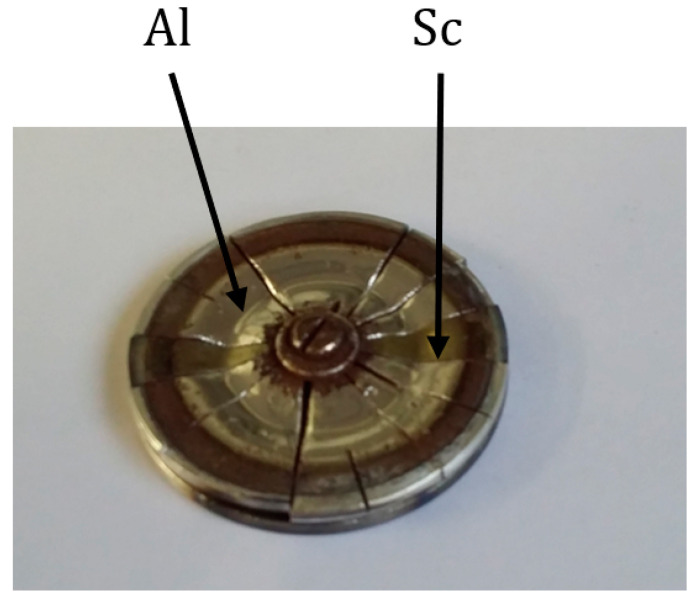
Picture of the sputtering target.

**Figure 2 sensors-20-04630-f002:**
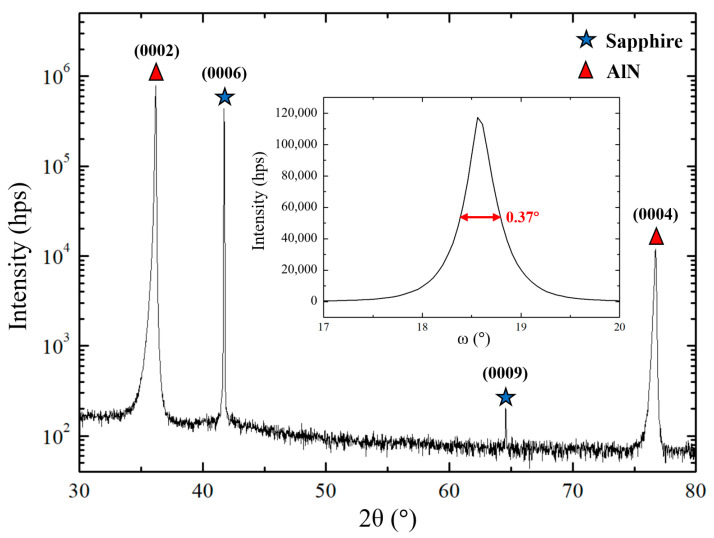
θ-2θ diagram of an AlN thin film deposited on a 470 °C heated sapphire substrate. The inset figure shows the rocking-curve of the (0002) orientation.

**Figure 3 sensors-20-04630-f003:**
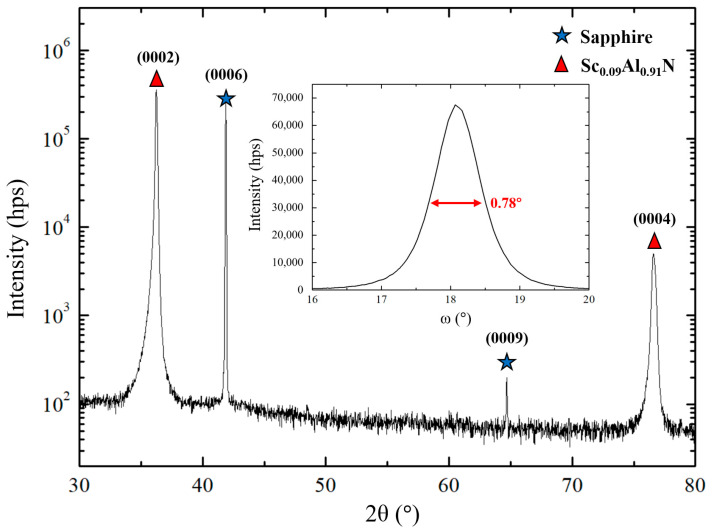
θ-2θ diagram of a Sc_0.09_Al_0.91_N thin film deposited on a 650 °C heated sapphire substrate, with a plasma pressure of 7 mTorr. The inset figure shows the rocking-curve of the (0002) orientation.

**Figure 4 sensors-20-04630-f004:**
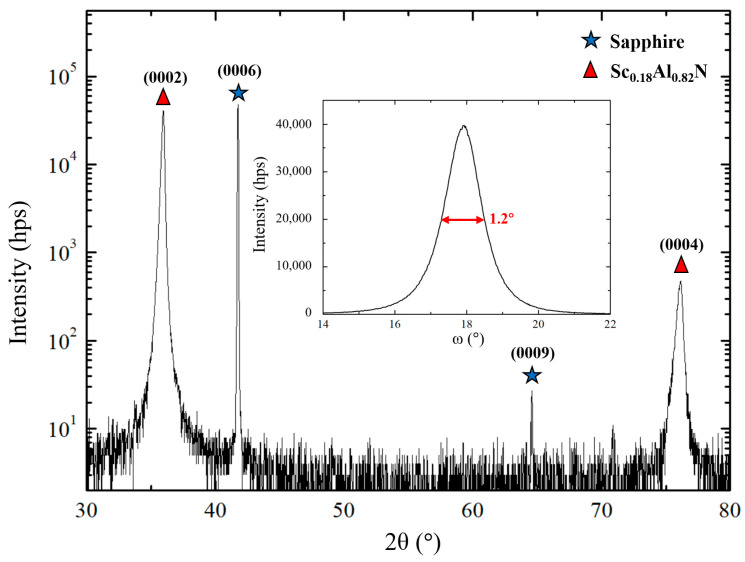
θ-2θ diagram of a Sc_0.18_Al_0.82_N thin film deposited on a 650 °C heated sapphire substrate, with a plasma pressure of 9 mTorr. The inset figure shows the rocking-curve of the (0002) orientation.

**Figure 5 sensors-20-04630-f005:**
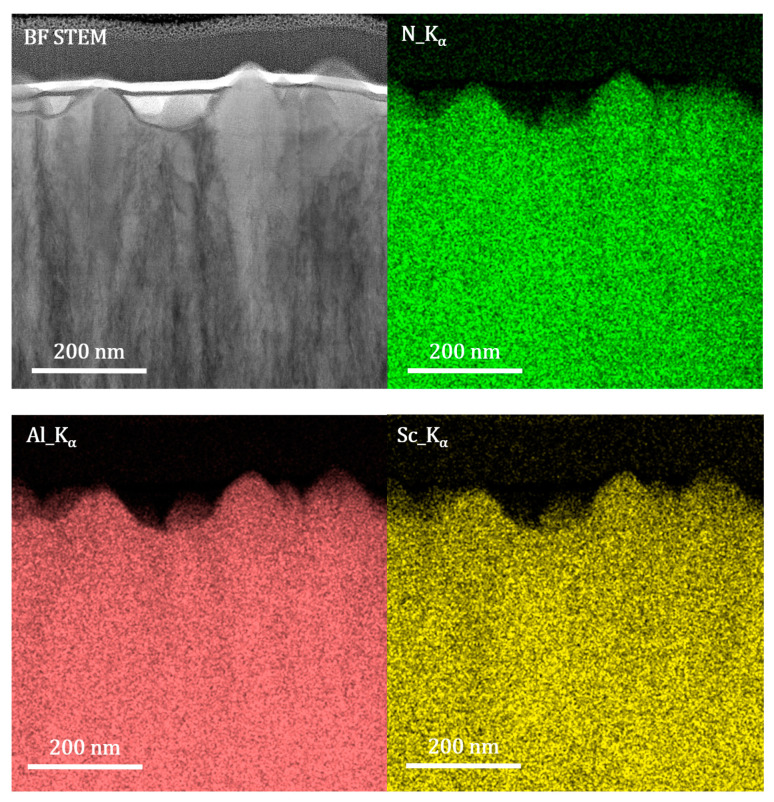
STEM-EDS mapping of the Sc_0.18_Al_0.82_N thin film: N (green), Al (red) and Sc (yellow).

**Figure 6 sensors-20-04630-f006:**
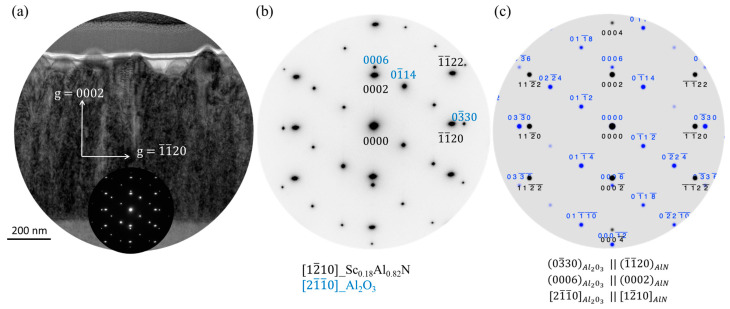
TEM bright-field image of a Sc_0.18_Al_0.82_N thin film deposited on a sapphire substrate (**a**). Composite SAED pattern recorded along the ${\left[1\bar{2}\bar{1}0\right]}_{Sc_{0.18}Al_{0.82}N}∥{\left[2\bar{1}\bar{1}0\right]}_{Al_{2}O_{3}}$ zone axis from the area on either side of the interface between the Sc_0.18_Al_0.82_N thin film and the Sapphire (**b**). Corresponding simulated diffraction pattern revealing the orientation relationship developed between the deposited thin film and the substrate (**c**).

**Figure 7 sensors-20-04630-f007:**
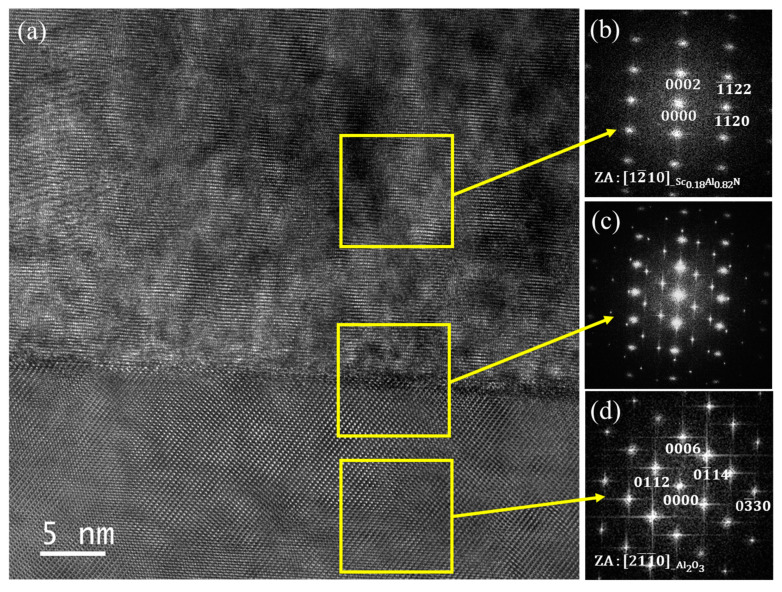
HRTEM micrograph made on the Sc_0.18_Al_0.82_N/sapphire structure (**a**) and corresponding FFT of the Sc_0.18_Al_0.82_N thin film (**b**), film/substrate interface (**c**) and the sapphire substrate (**d**).

**Figure 8 sensors-20-04630-f008:**
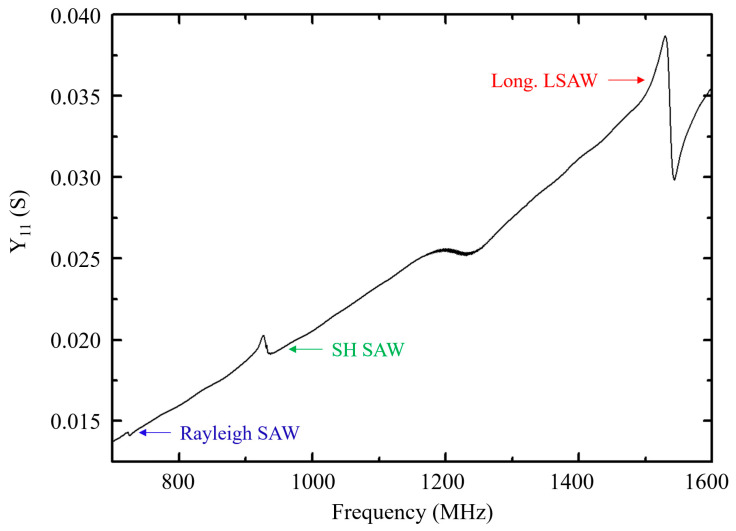
Y_11_ response of the SAW resonator based on the Sc_0.09_Al_0.91_N/sapphire bilayer structure, for a film thickness of 2.76 µm and a wavelength of 6.5 µm.

**Figure 9 sensors-20-04630-f009:**
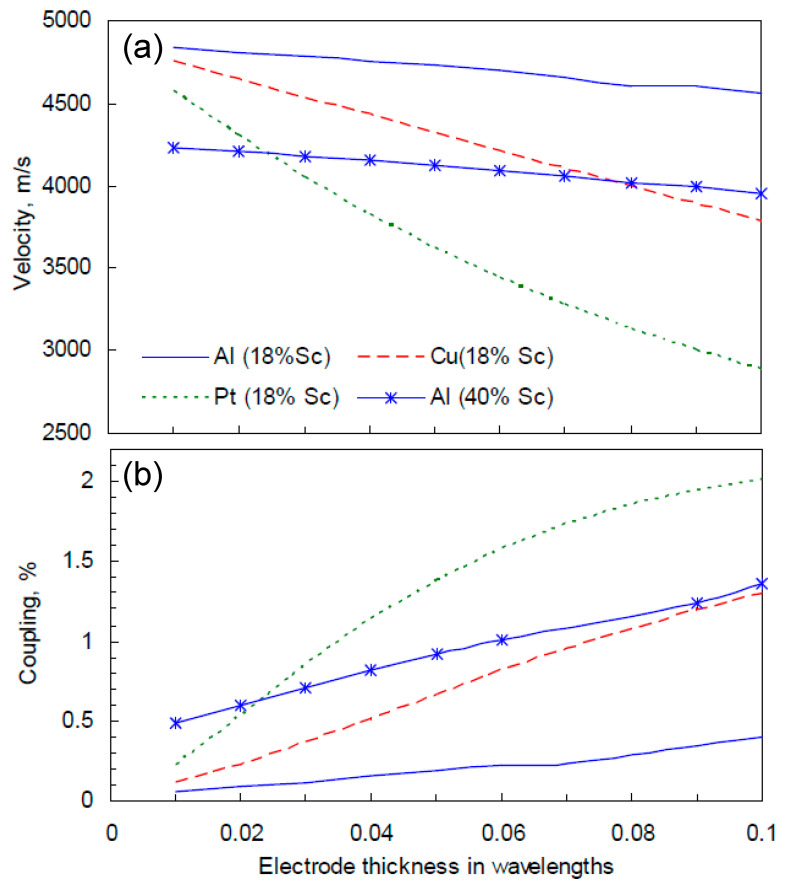
Simulated Rayleigh SAW velocities (**a**) and electromechanical coupling coefficients (**b**) in Sc_x_Al_1−x_N/sapphire with Al, Cu or Pt grating and different Sc contents (x = 0.18 and 0.4).

**Table 1 sensors-20-04630-t001:** Comparison between SAW properties of LGS, AlN/sapphire and Sc_0.09_Al_0.91_N/sapphire materials.

	SAW Velocity (ms^−1^)	K^2^ (%)	Acoustic Propagation Losses at 1 GHz (mdB/λ)
LGS (0°, 138.5°, 26.6°)	2700	0.4	4
(002) AlN/(006) Sapphire	5500	0.3	0.7
(002) Sc_0.09_Al_0.91_N/(006) Sapphire	10,000 (Longitudinal SAW)	1% (Longitudinal SAW)	Not known

**Table 2 sensors-20-04630-t002:** Growth conditions.

Target	Composite Target Sc (99.99%)/Al (99.99%)
Substrate	(006) Sapphire (dimension 20 mm × 15 mm)
Substrate temperature	Adjustable from the ambient to 740 °C
Sputtering Power	200 W
Target-Substrate distance	50 mm
Plasma pressure	Adjustable from 5 to 9 mTorr
Sputtering gas	99.999% N_2_

**Table 3 sensors-20-04630-t003:** Investigated sputtering parameters for each grown material.

Grown Material	Scandium Composition in the Composite Target (%)	Plasma Pressure (mTorr)	Substrate Temperatures (°C)
AlN	0	5	25, 290, 380, 470, 560
Sc_0.09_Al_0.91_N	12.5	5, 6, 7, 8	470, 560, 650, 740
Sc_0.18_Al_0.82_N	25	7, 8, 9	650, 740

**Table 4 sensors-20-04630-t004:** Space groups and lattice parameters of sapphire and AlN.

Material	Sapphire	AlN
Space group	$R\bar{3}c$	$P6_{3}\mathrm{mc}$
Lattice parameters	a = b = 4.7605 Å; c = 12.9956 Å α = β = 90°; γ = 120°	a = b = 3.084 Å; c = 4.948 Å α = β = 90°; γ = 120°

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
