# Peer review of "Epitaxial Growth of Sc0.09Al0.91N and Sc0.18Al0.82N Thin Films on Sapphire Substrates by Magnetron Sputtering for Surface Acoustic Waves Applications"

_sensors, 2020, doi:10.3390/s20164630_

Round 1
Reviewer 1 Report
The authors have described the effect of Sc on the growth of AlN film on Sapphire substrate prior to SAW applications, I recommend a publication in Sensors after a major revision because I strongly believe that this manuscript can be improved as well. The comments are listed below:
- Chemical formula in Title is not correct.
- The first 5 line of introduction is not clear to the readers because the authors described the literature poorly.
- The authors are advised to make a table with a comparison between langasite crystals, AlN and ScAlN on sapphire substrates, it will become more clear to readers
- The authors have mentioned “Sapphire substrates are highly compatible with AlN films as they show similar physical properties” but it is still unclear to readers what physical properties are.
- Methods: can the authors give more information of suppliers of their materials?
- Can the authors explain why θ-2θ and rocking-curve measurements are used in this work? Rocking curve measurements were made on (002) peak and it shows a FHWM value of 0.37, therefore it means it has an excellent out-of-plane texture. But can the authors explain why they performed rocking curve measurements on (002) peak and why they confirm it show an excellent FWHM value.
- Figure 2: Please remove “angle” from x-axis.
- How did the authors confirm that “the texture of the film was slightly worse compared to that of the optimized AlN films”
- Please change (Tab. II). to Table 2, please respect the author guidelines for submission.
- Figure 2 show XRD patterns of 470°C, while Figure 3 show XRD patterns of 650°C, but the authors has mentioned in the text that the temperature is raised from 560°C until 650°C. Therefore, where does this temperature of 560°C comes from? The authors also confirmed it is highly textured as FWHM is equal to 0.78. But as mentioned in previous comment: how did the authors confirmed that? It is also same for Figure 4 where the pressure is increased.
- If the authors mentioned the grain size is large, please provide the values. Indicating them in Figure 5.
- All of the figures in this manuscript show poor resolution. (A),(B), … caption should be in upper left corner position and not in random position.
- Figure 6, EDS mapping was performed, can the authors provide which K, L, M were used during EDS measurements.
- The authors has mentioned that FFT pattern shows only localized spots, which reveals that the film is epitaxy not only out-of-the plane, but also in-plane. How can the authors see this based on FFT pattern, it show in-plane epitaxial structure?
- It would be nice if the authors show the crystal structure of AlN and Sapphire to compare them with FFT patterns. What is a difference between SAED and FFT in this work?
Reviewer 2 Report
From my point of view, the "letter" format does not provide such a large amount of text and drawings. The article title is incorrect. It makes sense to change the form of presentation of the material. There is a base coating of aluminum nitride. There is an increase in the concentration of scandium in it. The emphasis should be shifted. Talk about AlN, AlSc0.09N, AlSc0.18N coatings! Authors need to do some serious work and fix a lot of small mistakes ("Scaln", "500. nm, (002)-oriented AlN", "omega/theta on XRD patterns", "Int/LnInt" etc.). The question of uniformity of coatings when using a mosaic target is not discussed. Whether the rotation of the substrate? Poor detailization of the deposition process in Exptl Chapter. Not enough articles by other authors on coatings in the Al-Sc-N system have been analyzed. In General, the article is of interest for publication.
Reviewer 3 Report
The paper presents the results of the growth of epitaxied AlN, Sc0.09Al0.91N and Sc0.18Al0.82N thin films on sapphire substrates.
Before the publication, it is worth to add some modifications.
In the abstract, the authors have written "the very good crystalline quality of the films at the interface" and "scandium can exhibit high piezoelectric properties" These parts are the opinion of the authors. In the abstract, facts and values should be written.
There are too few references from the last five years. Some additional studies on thin films should be described in the Introduction section. There is also too little information about SAW sensors.
Fig I and Tab I and II should be placed just after they appear in the text. The same suggestion is about other Figures.
The description of Table I consists of the sentence: "The variable parameters are in blue." - please clarify it.
There is a poor description of Fig 5 and 6. The sputtering process and simulation description is also insufficient. After reading the paper the reader can not repeat the experiment.
in the text there is the unit m.s-1. It should be ms-1.
Round 2
Reviewer 1 Report
The authors have improved the manuscript based on the comments of reviewers. However, I disagree with two comments concerning TEM analysis.
- The grain size was mentioned but how sure are the author it is grains and not twin boundaries? If this film is tilted in a specific diffraction zone, it will come clear which defects it are. Therefore, what is diffraction vector of this thin film analysis?
- EDS K, L, M were introduced but it is not mentioned which one (alpha, beta or?)
- Concerning SAEDP and FFT, the authors have explained what these techniques are, however, to be my honest, I’m familiar with these techniques. I had raised a question how the authors confirm that FFT show an in-plane and out-of-plane epitaxial structure?
Reviewer 2 Report
Now all the issues are addressed. I recommend to accept this paper.
Author Response
Thanks very much for your help in improving our manuscript.
Reviewer 3 Report
All my remarks have been taken into account. In my opinion, the paper is ready to be publish.
Author Response

(The authors gave the same response as above.)
